# Vitamin D from UV-Irradiated Mushrooms as a Way for Vitamin D Supplementation: A Systematic Review on Classic and Nonclassic Effects in Human and Animal Models

**DOI:** 10.3390/antiox12030736

**Published:** 2023-03-16

**Authors:** Mariangela Rondanelli, Alessia Moroni, Marco Zese, Clara Gasparri, Antonella Riva, Giovanna Petrangolini, Simone Perna, Giuseppe Mazzola

**Affiliations:** 1IRCCS Mondino Foundation, 27100 Pavia, Italy; 2Department of Public Health, Experimental and Forensic Medicine, Unit of Human and Clinical Nutrition, University of Pavia, 27100 Pavia, Italy; 3Endocrinology and Nutrition Unit, Azienda di Servizi Alla Persona ‘‘Istituto Santa Margherita’’, University of Pavia, 27100 Pavia, Italy; 4R&D Department, Indena SpA, 20139 Milan, Italy; 5Department of Food, Environmental and Nutritional Sciences, Division of Human Nutrition, Università Degli Studi di Milano, 20133 Milan, Italy

**Keywords:** vitamin D-enriched mushrooms, vitamin D-enhanced mushrooms, medical mushrooms, vitamin D, 25(OH)D, 25(OH)D2

## Abstract

Recent literature has shown that vitamin D, in addition to its well-known activity on the skeleton, has many positive effects on health. Unfortunately, it is not easy to meet intake needs solely with food. Mushrooms could provide a valid way to achieve this goal, because they are one of the few sources of vitamin D. The aim of this systematic review was to summarize what has been reported in the literature on the treatment of animal and human models with irradiated commercial mushrooms, with particular attention paid to the effects on clinical outcomes associated with the classical and nonclassical vitamin D functions. A total of 18 articles were selected. Six studies were conducted on human samples, while twelve were focused on animal models. The six studies conducted in humans involved a large number of subjects (663), but the treatment period was relatively short (1–6 months). Furthermore, the treatment dosage was different in the various groups (600–3800 IU/day). Probably for this reason, the studies did not demonstrate clinical efficacy on the parameters evaluated (cognitive functions, muscle system/function, metabolic syndrome). Indeed, those studies demonstrated an efficacy in increasing the blood levels of 25(OH)D2, but not in increasing the levels of 25(OH)D total. In 9 of 12 studies conducted on the animal model, however, a clinical efficacy on bone metabolism, inflammation, and cognitive performance was demonstrated. The results of this systematic review indicate that the intake of vitamin D from irradiated mushrooms could possibly help to meet vitamin D needs, but the dosage and the time of treatment tested need to be evaluated. Therefore, studies conducted in humans for longer periods than the studies carried out up to now are necessary, with defined dosages, in order to also evaluate the clinical efficacy demonstrated in animal models both for the classical (bone metabolism) and nonclassical (muscle function, cognitive performance, anti-inflammatory, and antioxidant activities) effects of vitamin D.

## 1. Introduction

The term mushrooms (or higher fungi) identify species ascribable to the kingdom of fungi that produce macroscopic fruiting bodies, mostly belonging to the phylum Basidiomycota and Ascomycota [1]. Edible mushrooms are generally characterized by a low caloric value, a good lipid profile (such as oleic, linoleic and linolenic acids), and a protein content (up to 44.93% by dry weight in dried foods). Mushrooms are also rich in some micronutrients (such as vitamin D) and bioactive compounds, such as beta glucans and antioxidant sterols [2,3]. For these reasons, together with an optimal content of umami compounds, such as glutamate, they are often highly valued and used in balanced dietary therapy programs. There are some differences between cultivated and wild mushrooms. For example, wild mushrooms generally contain higher contents of fiber and bioactive compounds than cultivated mushrooms [2], but they can accumulate higher amounts of heavy metals, such as arsenic, mercury, Ni, Cr, Pb, Cd and even radionuclides, especially if grown in polluted soils [2,4,5].

Vitamin D is an essential micronutrient for bone metabolism, as well as for the management of various clinical conditions and physiological metabolisms. Collectively, these particular vitamin functions are known as the nonskeletal (or nonclassical) functions of vitamin D. The main nonclassical functions of vitamin D are prevention and management of cancer, autoimmunity phenomena, viral infections, cardiovascular issues (including, e.g., acute myocardial infarctions and hypertension), metabolic syndromes and dysmetabolism, management of body weight, pain and neuromuscular function [6,7,8,9]. Ergocalciferol (vitamin D2) is an isoform of vitamin D used in fortified foods and dietary supplements [9]. It differs from the better-known D3 isoform (cholecalciferol) by having a double bond between carbons 22 and 23 of the polycyclic structure, and by the presence of a methyl group on carbon 24. Vitamin D2 precursors (known collectively as ergosterols) are capable of absorbing UV radiation between 240 and 320 nm, which mainly includes UVB rays. Irradiation with these rays induces the conversion of ergosterols to pre-vitamin D2 [1,10]. The activation pathway includes hydroxylations at C25 and C1alpha, producing final active form 1,25(OH)2D3. Alternative pathways of vitamin D2 activation are enhanced by CYP11A1, which produces biologically active hydroxyderivatives [11,12,13,14]. Similar activity of CYP11A1 is exerted on ergosterol [15,16].

As reported by several studies published in the literature, irradiation of fresh or dried cultivated mushrooms with ultraviolet rays or sunlight generally leads to a significant enrichment of vitamin D2, inducing the conversion of ergosterols to ergocalciferol. This occurs to a very important extent in commercial mushrooms more so than in wild mushrooms, as most cultivation techniques of these organic matrixes take place predominantly in the dark [1,17]. According to the literature, about 10 species are regularly cultivated commercially [18]. Only three species (Agaricus bisporus, Pleurotus oyster mushrooms and Lentinula edodes) account for about three quarters of all mushrooms consumed [1,17,19,20,21].

Published studies [1,22,23,24,25,26,27,28,29] have identified several technological and methodological variables that can influence the yields of this reaction, including the species of mushroom considered, the type of ultraviolet ray used, the size of the sample (and consequently the irradiation surface area and the number of sample surfaces exposed to radiation), the density of irradiation, and its duration. Furthermore, it appears that factors unrelated to the reaction conditions may also exert a non-negligible effect on the final vitamin D2 content in these enriched mushrooms, including the type of product treatment, the storage temperature, and the way the sample is cooked. A synopsis of the latest systematic literature reviews on the effect of these technological and environmental factors is presented in Table 1.

Consuming vitamin D-enriched commercial mushrooms could be a strategy to maintain adequate levels of vitamin D intake, especially considering the rarity of finding the true source of this important vitamin in food, such as fish liver [1]. However, the clinical and nutritional use of ergocalciferol still raises some doubts: although ergocalciferol is apparently absorbed with similar efficiency to cholecalciferol, it is less bioactive than the D3 isoform, both in the case of the classical and nonclassical functions [6,7,8]. Still, recent evidence shows that ergocalciferol is subject to greater losses during food storage than cholecalciferol and is generally more susceptible to degradation with cooking [30,31,32,33]. Moreover, according to a recent study published in the literature, irradiation of commercial mushrooms seems to induce changes not only in the vitamin D2 content, but also in the relative concentrations of some nutritionally interesting molecules, including some polysaccharides and phenolic compounds, which influence the antioxidant power (DPPH radical scavenging activity) of the analyzed samples [34]. Irradiation with ultraviolet rays, on the whole, seems to induce an increase in the concentration of phenolic compounds, which, in the study by Kido et al., seems to be able to “compensate” for the decrease in antioxidant power resulting from hot drying mushrooms, which generally decreases the antioxidant power of their aqueous extracts [34].

The health effects related to chronic subclinical inflammation are not only related to antioxidant species but to an overall management of oxidative stress, which is also related to an optimal function of the immune system, and in this context, vitamin D plays a central role [35]. For this reason, the relative concentration of these compounds obtained through these treatments, together with the increase in ergocalciferol, could induce a positive effect on the management of chronic inflammatory phenomena resulting from the intake of these mushrooms. This potential effect could be useful both in the context of animal nutrition (improving the nutritional profile of food fed with enriched mushrooms) and in human nutrition, specifically acting on those clinical outcomes related to the classical and nonclassical vitamin D functions, similar to what happens in other foods generally considered “healthy”, such as olive oil [22]. Nevertheless, the evidence is still preliminary and data are lacking to support a significant positive effect associated with the consumption of vitamin D-enriched mushrooms on clinical outcomes, both in animal models and in humans. We still do not precisely know the factors capable of maximizing these presumed positive effects, both with regard to many of the technological and environmental aspects summarized in Table 1, and to clinical aspects such as the doses and modes of consumption, the period of treatment and possible genetic, epigenetic and environmental factors. The latter of those could justify the implementation of these mushrooms in specific personalized diet therapies and in the context of precision medicine, both as a dietary supplementation and as a food in itself.

The aim of this systematic review was to summarize what has been reported in the literature on the treatment of animal and human models with irradiated commercial mushrooms. Special attention is paid to the effects of treatment on clinical outcomes associated with the classical and nonclassical vitamin D functions since, to the authors’ knowledge, there are no systematic reviews or meta-analysis concerning the topic. In particular, the main objective was to investigate and thus understand the state of the art of nonclassical functions/effects of vitamin D enriched mushrooms, either in human or animal models, in order to point out the beneficial effects of vitamin D on the organism as a whole, not just the well-known and documented classical effects of vitamin D.

## 2. Materials and Methods

The research was performed on PUBMED, searching for articles published before 20 October 2022. We made the application process on PROSPERO—International prospective register of systematic reviews. The acknowledgment of receipt number is 402641. The authors used a structured research strategy (Appendix A). The selection of the studies was conducted independently by two authors (G.M. and A.M.), first with a screening of titles and abstracts, and then a second full-text review of the studies found to be potentially relevant. In case of disagreement between the two, a third person intervened (M.Z.) to resolve the dispute. All information regarding the identified papers, abstracts, and full-text articles screened, as well as excluded articles and those included in the qualitative and quantitative synthesis, are shown in Figure 1 and in the PRISMA diagram (Figure 1, [31]). In the absence of specific Mesh Terms dedicated to enriched mushrooms, the research was conducted by carrying out the Bolerian questions set out in Appendix A. Selected in a first phase (PHASE 1) were all published studies conducted on a total sample consisting of 10 of the best-selling commercial mushrooms [18] treated with UV irradiation or sun exposure, net of all technological, environmental, and clinical variables. The studies included were case-control studies (RCTs and studies with a control group), conducted in animal models and humans. Only samples treated with mushrooms directly irradiated with UV light or exposed to direct sunlight for a certain period, either as fresh food, or dried and/or powdered (STEP 2), are included the review.

This subdivision of work was necessary in order to include all studies in the interrogated database. From the final selection of data, case reports and studies not published in English were therefore excluded. Again, guidelines and gray literature were excluded.

## 3. Results

Overall, a total of 18 articles were selected for analysis. Six studies were led on human samples (Table 2) while 12 focused on animal models (Table 3). Within each sample’s category, the heterogeneity of protocol and study design was underlined.

### 3.1. Studies Conducted on Human Samples

Concerning the classical functions of vitamin D (effects on bone), none of the papers analyzed had directly investigated BMD (Bone Mineral Density) values. Conversely, all the studies analyzed how vitamin D from mushrooms could increase precursors 25(OH)D2, 25(OH)D3 or total serum levels. Five out of the six studies conducted on human samples have mainly examined the nonclassical function of vitamin D, such as effects on inflammation regarding muscles, cognitive performance and mood, as well as metabolic syndrome.

Shanely and collaborators (2014) [41] conducted a randomized, double-blind study on a sample of 33 athletes (males, 16.2 ± 0.19 years) with a low serum 25(OH)D level (<30 ng/mL). The authors investigated whether the intake of vitamin D2 from portobello mushroom powder could induce an improvement in skeletal muscle function as well as attenuate exercise-induced muscle damage. Participants were randomized into groups with vitamin D or placebo and received a 6-week supply of capsules. The experiment involved the ingestion of two capsules per day (each containing 300 IU of vitamin D2, derived from vitamin D-enriched portobello mushrooms and dried magnesium stearate). The placebo-treated participants, on the other hand, ingested capsules that were entirely similar to the treated group, but differed only in their vitamin D content, being composed of powdered portobello mushrooms not exposed to UV light and magnesium stearate. The results showed that in the group treated with the enriched mushrooms, there was a statistically significant increase in 25(OH)D2 levels (*p* < 0.0001), a decrease in 25(OH)D3 levels (*p* < 0.0001), and an increase in serum total 25(OH)D concentration (*p* < 0.01). From the results obtained, it was inferred that muscle function in the treated group did not differ significantly from that estimated in the placebo group. Again, similar values of serum MB (Serum myoglobin) (interaction effect, *p* = 0.927), LDH (lactate dehydrogenase) (interaction effect, *p* = 0.465), AST (aspartate aminotransferase) (interaction effect, *p* = 0.673) and total CPK (creatine phosphokinase) (interaction effect, *p* = 0.469) were found in the treated groups. From the results presented, the authors therefore concluded that vitamin D2 supplementation for six weeks (600 IU/day) was directly associated with a significant increase in serum 25(OH)D2 levels of 9.3 ng/mL, but this increase did not have a significant effect on markers of muscle function, nor did it attenuate exercise-induced muscle damage or delayed onset of muscle soreness (DOMS).

Nieman and collaborators (2013) (Nieman et al., 2013) [43] conducted a trial on a sample of 28 US athletes in North Carolina. The aim of the study was to investigate the effect of a six-week supplementation with vitamin D2 (vitD2, 3800 IU/day) on muscle function, as estimated by evaluating the effect of this protocol on exercise-induced eccentric muscle damage (EIMD) and delayed onset of muscle soreness (DOMS). Subjects were randomized into two groups: a group treated with 3800 IU/day of vitamin D2 obtained from fresh portobello mushroom powder enriched in vitamin D (*n* = 13) and a placebo group (*n* = 15). Supplementation in the active group did not induce significant changes in the final levels of total 25(OH)D (*p* = 0.127), although a significant increase in the serum 25(OH)D2 levels of approx. 14 ± 1.96 ng/mL (*p* < 0.001), as well as a decrease in serum 25(OH)D3 values of approximately −7.48 ± 2.28 ng/mL (*p* = 0.036) compared to placebo, in which there were changes of 0.076 ± 1.19 ng/mL in serum 25(OH)D2 values and −2.11 ± 1.09 ng/mL in serum 25(OH)D3 values, respectively. It should be noted that the serum 25(OH)D3 levels were higher in the placebo group in October and increasingly lower as the months progressed, reaching their lowest values in December and January. With regard to muscle function, a statistically significant increase in serum myoglobin, LDH, CK, and DOMS values was found following eccentric training sessions after the end of the experimental protocol (*p* < 0.001); with significantly greater post-exercise serum myoglobin and CK levels in the vitamin D2 group compared to the placebo group (both interaction effects, *p* < 0.001). In contrast, the pattern of change in DOMS was not found to be statistically different between the groups (interaction effect, *p* = 0.490). The authors concluded that the treatment tested induced an increase in serum 25(OH)D2 values, but in the treated group, there was a decrease in serum 25(OH)D3 values and generally a worsening of exercise-induced muscle damage (EIMD). With regard to muscle function test scores, there were no statistically significant differences between the treated group and the placebo group.

A study conducted in Australia (Zajac et al., 2020 [38]) on healthy male and female subjects aged 60–90 years old (N = 436) investigated the effect of an intervention protocol consisting of four different treatments on participants’ serum vitamin metabolites, cognitive performance, and mood. The subjects were randomized into four experimental groups, the first consisting of subjects treated with mushrooms enriched in vitamin D2 (D2M) of approximately 600 IU, the second consisting of subjects treated with vitamin D3 supplementation of approximately 600 IU (D3), the third consisting of subjects treated with mushrooms not enriched in vitamin D2 (standard mushrooms, SM), and finally a fourth and final group was treated with a placebo (PL). Overall, the experimental results suggested a decrease in the mean levels of total 25(OH)Dtot over the 24-week period (all *p* ≤ 0.01) in subjects belonging to groups 1, 3, and 4, with the placebo group having the greatest effect and the decrease in 25(OH)Dtot values in the first group being less pronounced and slower (*p* = 0.047). In contrast, in the third group (i.e., the group treated with vitamin D3), there was a general increase in total 25(OH)Dtot values, (*p* < 0.001). Similarly, 25(OH)D3 values decreased from the baseline with a more pronounced and more rapid decrease in the first group, compared to the placebo group (*p* < 0.001). The 25-OH-D2 levels were generally not detectable at the baseline in any group. However, the number of individuals with detectable levels (>5.0 nM) was higher at the baseline in the first experimental group, i.e., the D2M group (*p* < 0.001). Furthermore, the levels of total 25(OH)Dtot were only statistically higher in the second experimental group (i.e., the one treated with vitamin D3), but this increase was only significant in those subjects with values below 20 ng/mL already at the baseline. The rate of decline in total 25(OH)D was found to be variable between experimental groups and greater in subjects with sufficient values of total 25(OH)D at the baseline. Finally, 25(OH)D3 levels were basically unchanged in the vitamin D3-treated subjects and generally decreased in the remaining experimental groups (D2M, SM and PL), with significant changes in each of the three groups just mentioned also compared to the baseline. With regard to the results on markers of cognitive function and mood, the overall time effects showed a general improvement in all domains except for memory quality, but the improvements induced were not generally traced back to the treatment, being similar between the treated subjects and the placebo group. In the analysis of the experimental data, generally positive effects were found for perceived stress levels and no tests measuring mood. Specifically, in the analysis of all experimental results relating to mood, only the three-way interaction between happiness and basal vitamin D level exceeded the criteria (*p* = 0.009). In summary, the results showed no significant effects of the studied treatments on total 25(OH)D values and on markers of cognitive function or mood in elderly subjects.

In the US study conducted by Mehrotra A et al. (2014) [39], conducted on 36 prediabetic adults with BMI > 25 and vitamin D deficiency (25(OH)Dtot ≤ 20 ng/mL), the researchers investigated the effects of EM supplementation on circulating concentrations of total 25(OH)D, 25(OH)D2, and 25(OH)D3 on metabolic syndrome risk factors. The study involved treatment with four different protocols: the first group involved supplementation with capsules containing UV-treated mushrooms containing 600 IU of D2/day and supplementation with placebo capsules; the second experimental group involved supplementation with a UV-treated mushroom containing approx. 4000 IU of D2/day and supplementation with placebo capsules; the third group involved supplementation with untreated mushrooms and food supplements labeled as containing 600 IU of cholecalciferol (D3); finally, the fourth group involved supplementation with capsules containing untreated mushrooms and food supplements labeled as having a total intake of 4000 IU of D3. From the experimental results, it can be seen that the treatment with enriched mushrooms had similar outcomes to other studies. In the second experimental group, treatment with UVB-enriched mushrooms resulted in a small but significant increase in serum 25(OH)D2 over time (*p* < 0.05), which, however, was not such as to lead to significant changes in total 25(OH)D values. Finally, in this experimental group, 25(OH)D3 values remained unchanged. In contrast, no significant changes in the blood levels of 25(OH)D2 and D3 were found in the third group. In the case of subjects treated with D3 supplementation at both dosages tested, the treatment was associated with a positive change in serum 25(OH)D3, with no change in 25(OH)D2. This increase was sufficient to induce a significant increase in blood levels of 25(OH)Dtot. Finally, in none of the experimental groups was a positive result found with regard to the secondary outcomes tested. Consequently, the study authors conclude that the treatment studied did not induce positive results on the metabolic syndrome markers tested.

Stepien et al. (2013) [42] conducted a 4-week study of 50 healthy Caucasian men and women living in and around Dublin (40–65 years). The aim of the study was to investigate possible differences between the consumption of 15 μg/day of vitamin D2-enriched mushrooms and a vitamin D3 supplementation protocol on hematochemical vitamin D markers. Participants were divided into four experimental groups treated, respectively, with 15 μg of vitamin D2-enriched mushroom powder or unenriched mushrooms and 15 μg of vitamin D3 or placebo capsules. Comparison of concentrations between the experimental groups showed that the concentration of 25(OH)D2 after the intervention was significantly higher in the group taking enriched mushrooms than in the group taking placebo mushrooms (*p* < 0.01). However, when comparing these two groups, the researchers found no significant changes in serum 25(OH)D3 and total 25(OH)D. Still, the experimental data suggest that the highest levels of total 25(OH)D found in all experimental groups was that obtained from the vitamin D3 capsule supplementation protocols, in which a statistically significant increase in 25(OH)D3 values was also found. With regard to the other outcomes investigated, a decrease in plasma values of PAI-1 (plasminogen activator inhibitor-1) was found in the group treated with vitamin D2-enriched mushrooms (*p* < 0.05). In conclusion, the results of the study indicate that supplementation with enriched mushrooms induces an increase in the serum values of 25(OH)D2, but not in the final values of total 25(OH)D.

Stephensen et al. (2012) [40] conducted a 6-week study on healthy adult subjects (*n* = 40; age: 20–59) recruited at the University of California, with the aim of investigating the bioavailability of ergocalciferol obtained from the consumption of Agaricus bisporus. The intervention was conducted in four experimental groups: a control group (C) that was treated with a protocol involving supplementation with untreated mushrooms, which provided 0.85 µg/d of ergocalciferol (*n* = 10); two groups (M1 and M2) whose participants were subjected to a supplementation protocol with UVB-irradiated (EM) mushrooms, such that they provided 8.8 (*n* = 10) and 17.1 µg/d (*n* = 9), respectively; and a further supplementation group (S) that was treated with purified ergocalciferol plus untreated mushrooms such that a total of 28.2 µg/d of vitamin D2 (*n* = 9) was provided. In the treated groups, serum 25(OH)D2 concentrations were low at the baseline, ranging from 0.6 to 8.7 nmol/L (*n* = 38), with no statistically significant differences between treatment groups or cohorts. EM mushroom consumption resulted in an increase in serum 25(OH)D2 in the M2 and M1 groups at 3 and 6 weeks, respectively. These increases were greater than in group C and generally similar between groups M2 and M1. The ematic values of 25(OH)D2 remained generally low in control group C throughout the experimental protocol. The group with the highest vitamin D2 intake, i.e., group S, was the one in which the highest mean 25(OH)D2 levels were recorded, both at 3 and 6 weeks, compared to all other experimental groups. Again, the experimental results show that the mean values of total serum 25(OH)D decreased significantly in the M2 and M1 groups, but not in the S or C groups. However, the mean values of 25(OH)D in the M2 and M1 groups on days 3 and 6 were similar to those found in the C or S groups. Again, the mean change did not differ from the baseline in the experimental groups on days 3 and 6 either. Thus, compared to group C, the treatments induced a significant increase in 25(OH)D2 levels compared to all other groups at both 3 and 6 weeks, but the supplementation protocols tested for groups M1, M2, and S generally induced no positive effect on overall vitamin D status. While vitamin D2 intake in the treatment groups induced an increase in 25(OH)D2 concentrations compared to control group C at weeks 3 and 6, the total 25(OH)D concentrations were not found to be different in the treatment groups and group C at either time point. These results suggest that ergocalciferol intake may cause a significant decrease in serum 25(OH)D3 values. On days 3 and 6, the mean 25(OH)D3 values in group C did not differ from the baseline. However, the mean serum 25(OH)D3 concentrations were lower in the S and M2 groups from week 0 to week 3 and in all three treatment groups from week 0 to week 6. Overall, the experimental results suggest that ergocalciferol from UVB irradiation of mushrooms is bioavailable. However, the serum levels of 25(OH)D3 decreased in proportion to the increase in 25(OH)D2, and this decrease was such that the researchers observed no improvement in overall vitamin D status in the experimental groups tested.

### 3.2. Studies Conducted on Animal Samples

The studies analyzed were mainly conducted in mice or pigs models, ranging from 30-subject samples to 300 (gender depending on study). All the studies concerned effects of EM mainly on bone health, thus focusing on the classical function of vitamin D. Only six studies investigated the effects of EM on the nonclassical function of vitamin D, taking into consideration outcomes such as immune system, microbiology, volatile fatty acids, coefficient of apparent total tract digestibility (CATTD), gastrointestinal morphology, and nutrient transporter genes.

A 10-week study was conducted in Canada (Calvo et al., 2012 [44]) on 300 female Sprague–Dawley rats weaned and fed five diets. The vitamin D-deficient and control diets contained no mushroom powder that provided the recommended amount of vitamin D3 in one arm and were completely devoid of vitamin D3 in the other arm, while the other test diets were all designed to contain a vitamin D-free mixture to which either 5.0% unexposed mushroom powder or 2.5% or 5.0% UVB-treated EM powder was added at the expense of maize starch. From the results, the researchers found higher PTH levels in the vitamin D-deficient fed control rats compared to the vitamin D supplemented groups (*p* < 0.001). The mean PTH levels in the vitamin D-deficient mushroom diet rats were also significantly lower (*p* < 0.002) than in the vitamin D-deficient control group, suggesting that these food matrixes may contain factors that can positively influence calcium absorption and thus reduce PTH production. Rats fed the 5.0 percent EM diet had a mean plasma 25(OH)Dtot level of 159 ± 29 ng/mL, while the control group had the mean final 25(OH)Dtot values of 32 ± 11 ng/mL. Rats fed the 5.0 percent mushroom diet showed statistically significant (*p* < 0.01) mean connective tissue density and trabecular mineralization but lower trabecular thickness than the control group. Consequently, the researchers concluded that mushroom treatment of rats could stimulate bone growth, both in the case of enriched and unenriched mushrooms, as suggested by the longer femurs in mushroom-treated animals (*p* < 0.001). Nevertheless, a higher vitamin D intake resulting from the experimental protocols characterized by the consumption of EM mushrooms induced a greater effect on cortical bone, as indicated by the increased mean cortical tree thickness and pMOI (*p* < 0.01) obtained in rats fed EM mushrooms compared to rats fed non-enriched mushrooms. However, no differences were observed in cortical bone strength values based on load-to-break measurements. Furthermore, plasma creatine levels were not dissimilar between the groups, suggesting that treatment with these experimental protocols has no adverse effects on renal function, even considering the high dietary intake of vitamin D and the high hematochemical values of 25(OH)Dtot achieved. The authors therefore concluded that vitamin D2 from UVB-irradiated mushrooms is easily absorbed and acts adequately in the bone. Furthermore, similar supplementation protocols in the animals tested induced no significant adverse effects on kidney function and was instead effective in supporting bone growth and mineralization in growing mouse models.

Jasinghe et al. (2006) [45] conducted a study on 30 male rats to investigate the effect of consuming shiitake EM, (containing 1 mg D2/d) on certain parameters related to vitamin D bone function and body weight. The results showed that body weight at the beginning and end of the study did not differ between the groups, nor did bone length of the femur, but the BMD of the femur in group 2 was significantly higher (*p* < 0.01) than that of the other two groups. Furthermore, the BMD values of the right and left femurs within the groups did not differ statistically significantly. The serum concentration of 25(OH)Dtot in group 2 was 129 ± 42 (SD 22 ± 00) nmol/L, which was significantly different from that of groups 1 and 3; specifically, in group 3 the values found were 6 ± 06 (SD 1 ± 09) nmol/L and were associated with a decrease in 25(OH)Dtot concentration compared to group 1. In contrast, a significant increase in 25(OH)D was observed in group 2. Serum Ca levels in group 2 were significantly lower than in group 1, probably due to a higher rate of bone mineralization in group 2 (which received vitamin D2 from mushrooms) than in group 1. This is supported by the observation of significantly higher BMD and femur length in group 2. In addition, decreased serum levels of parathyroid hormone, increased serum levels of ionized Ca, and an age-related decrease in duodenal Ca uptake have been reported to contribute to this difference. The authors concluded that EM vitamin D2 is generally bioavailable and able to induce a significant increase in femoral bone mineralization (*p* < 0.01) compared to controls.

Shin-Yu Chen and collaborators (2015) [53] performed a 23-week intervention in 32 7-week-old female mice (mouse model of osteoporosis). The sample was divided into 4 groups: (1) Bilateral ovariectomy (OVX) was performed on 24 mice. (2) A further 8 mice underwent sham surgery. The 24 OVX mice were randomly assigned to one of the following groups: (3) OVX (no treatment) (9 mice); PM (receive UV-PM) (8 mice); NPM (receive no UV-PM) (7 mice). The authors wanted to investigate the role of EM in indicators of osteoblasts and osteoclasts (bone density) as well as bone metabolites. The results showed that the OVX, NPM, and PM mice had higher body weights than the sham mice from week 17 to the end of the experiment. The measurements of Tb.BMD, BV/TV, BS/TV, Tb.Th, Tb.N, and Conn.Dn of the PM mice were greater relative to the measurements in the OVX mice, while the measurements of BS/BV and Tb.Pf of the PM mice were lower relative to the measurements in the OVX mice. The NPM mice also showed significantly higher Tb.BMD, BV/TV, and Tb.Th values and a lower BS/BV ratio than the OVX mice. Comparison of bone parameters between NPM and PM mice revealed that BS/BV and Tb.Pf of PM mice were lower than those of NPM mice. The comparison between sham and OVX mice showed no differences in the mean diaphysis of the femur. The EC, PC, and cross-sectional area of the mid-femoral diaphysis of NPM mice were higher than those of OVX mice, but the thickness of the mid-femoral diaphysis did not differ between treatments. At week 12, osteocalcin levels were increased in PM mice compared to OVX and NPM mice, suggesting that increased bone formation may have occurred in PM-fed mice at this time and in this experimental group. At week 12, the levels of ALP were higher in PM mice than in OVX mice, but this trend reversed at week 23, with significant decreases in the levels of ALP in PM mice compared to OVX mice. Both PYD and NTXI (biomarkers of bone resorption rate) levels were lower in PM mice than OVX mice at 23 weeks. Again, calcaemia was lower in PM mice compared to sham and NPM mice, in contrast to phosphataemia, which was similar in both groups. In addition, PM mice had significantly higher levels of amino acids and metabolites related to energy metabolism than OVX mice, including taurine. Eighteen metabolites were primarily responsible for the observed difference between the NPM and PM treatment groups at week 23. They showed significantly higher levels of amino acids and energy metabolites and lower levels of 3-hydroxybutyrate and metabolites related to the gut microbiome in PM mice compared to NPM mice. The authors concluded that treatment with P. ferulae enriched with vitamin D2 by UV irradiation could have a positive effect on bone mineralization, as well as a decrease in markers related to bone resorption and generally an increase in metabolites related to overall bone health. These positive effects were attributed by the researchers to the polyphenol and fiber content of *P. ferulae*, as well as the high vitamin D2 content.

Lee et al. (2009) [51] conducted a study on 55 osteoporotic mice randomly divided into 11 groups (*n* = 5 for each group in the same cage), all fed a low-calcium and vitamin D3 diet for a period of 4 and 8 weeks of age. As a positive control, one group was fed a normal diet for the same time period. To investigate the effect of vitamin D2-enriched mushrooms on bone mineralization and the regulation of active calcium transport gene expression, the diets of 9 of the low-calcium and vitamin D3 groups were supplemented with 5, 10 and 20 per cent enriched mushroom powder for 4 weeks, a supplementation protocol that provided 0.5 µg, 1 µg and 2 µg of vitamin D2, respectively. In this study, although the authors did not measure 25(OH)Dtot levels, they found a general improvement in bone thickness, femoral density, and serum calcium levels in the treated mushrooms. Again, the results of this study show that mRNA expression of duodenal TRPV6 and renal TRPV5 and TRPV6 (calcium channels located in the apical membranes of intestinal and renal epithelial cells and implicated in calcium absorption during transcellular calcium transport) increased in mice whose diet was supplemented with vitamin D2 and vitamin D-enriched mushrooms to the other diet groups. Again, the mRNA levels of duodenal and renal CaBP-9k (a cytosolic protein that has a high affinity for calcium ions) were regulated by vitamin D2 derived from enhanced oak mushrooms. In rodents, intestinal CaBP-9k is involved in intestinal calcium absorption and is transcriptionally and post-transcriptionally regulated by 1,25-(OH)2D; it was also shown that duodenal CaBP-9k expression can be linked to 1,25-dihydroxycholecalciferol in humans. Together with the results of the TRPV gene expression analysis, these results indicate that vitamin D2 derived from improved oak mushrooms is available in a bioactive form in mice. From the experimental results, the authors concluded that mushroom-derived vitamin D2 was able to induce the transcription of TRPV5 and TRPV6, promoting bone mineralization. Moreover, the increase in active calcium reabsorption in the kidney and duodenum also partly explains the increase in serum calcium levels in mice fed the highest vitamin D2 content. To summarize, the beneficial effect of vitamin D2 and vitamin D-enriched mushrooms on osteoporosis symptoms was assessed on the basis of femur density and length, bone histology, serum calcium levels, and mRNA levels of active calcium transport genes. Femur density, tibia thickness, serum calcium levels, and mRNA expression of active calcium transport genes were significantly higher in mice fed a low-calcium, vitamin D3-deficient diet, supplemented with vitamin D2 and calcium-enhanced oak mushrooms, compared to mice that did not receive supplementation. These results indicate that vitamin D2 and/or mushroom-derived calcium could improve bone mineralization directly as well as calcium absorption in the duodenum and kidney.

Dong Jae Won and collaborators (2019) [52] conducted a 6-week study on 48 female Sprague–Dawley rats with the aim of analyzing the bioavailability and bone loss inhibition effects of vitamin D2 derived from UV-irradiated shiitake mushrooms. Rats underwent sham operation or bilateral ovariectomy at the age of 5 weeks. The sample was divided into 6 groups: 3 sham and 3 ovariectomized (OVX) groups, i.e., control: vitamin D-deficient diets only; UV(X): vitamin D-deficient diets and non-irradiated mushroom powder; UV(O): vitamin D-deficient diets and irradiated mushroom powder. Comparison of the 25(OH)D2 levels between the sham UV(O) and OVX-UV (O) groups showed that the levels of this marker decreased by 70% in the absence of estrogen, suggesting that the presence of ovaries is highly associated with vitamin D2 bioavailability. In contrast, there was a decrease in both the sham UV(O) group and the OVX-UV (O) group compared to the groups that consumed a low vitamin D2 diet (Con), suggesting that the increase in 25(OH)D2 levels caused the onset of an antagonism phenomenon between 25(OH)D2 and 25(OH)D3. Interestingly, calcaemia levels were lower in the UV(O) groups than in the other treated groups, although there was also a decrease, albeit more modest, in calcaemia levels in the OVX groups. Again, the BMD values of the sham UV(O) group were found to be higher (*p* < 0.05) than the values obtained in the analyses of the sham-con and sham UV(X) groups. Taken together, these experimental results suggest that vitamin D2 from UV-irradiated shiitake mushrooms is bioavailable in these mouse models, as well as exerting positive effects on trabecular bone architecture, with BMD and Tb.Th values being higher in both the sham and OVX groups than in the control groups.

Duffy et al. (2018) [55] conducted a controlled longitudinal study in Ireland on 120 pigs (60 males, mean initial live weight: 58.0 kg, SD 4.6) to analyze total vitamin D activity in pork, levels of individual vitamins and their 25-hydroxymetabolites, meat quality, and antioxidant status. Pigs treated with enriched mushrooms had a higher final body weight (*p* < 0.01), an increased average daily gain (ADG) (*p* < 0.01), a better feed conversion ratio (FCR) (*p* < 0.01), and an increased carcass weight (*p* < 0.05) compared to pigs offered the other feed treatments during the experiment (day 0–55). Interestingly, final body weight, ADG, and FCR in pigs treated with vitamin D3 were similar (*p* > 0.05). Again, pigs fed vitamin D2 and vitamin D2-enriched mushrooms had a greater back fat depth (*p* < 0.05) compared to the vitamin D3 treatment.

Drori et al. (2017) [54] conducted a 25-week controlled longitudinal study in Israel involving a sample of 10-week-old male C57BL/6 mice. The experimental design comprised 4 groups (*n* = 6 each) all treated three times a week for 25 weeks with one of the following: Group A (control), saline (0.9% NaCl); Group B (vitamin D), 25 μL of an over-the-counter vitamin D supplement containing 400 IU per drop diluted in olive oil, equal to 10 IU per mouse per feed; Group C (LE), 25 μL of an LE mushroom extract containing 8.3 mg of dried mushrooms suspended in double-distilled water (DDW), per mouse and feed; Group D (LE + Vitamin D), 25 μL of vitamin D-enriched LE mushroom extract containing 8.3 mg of dried mushrooms and 10 IU of vitamin D suspended in DDW, per mouse and feed. In group C, there was a decrease in body fat accumulation and a decrease in hepatic fat, mined by means of magnetic resonance imaging. Again, in this experimental group there was a marked decrease in serum ALT and AST levels (from 900 and 1021 U/L in the control group to 313 and 340; 294 and 292; and 366 and 321 U/L for ALT and AST in the vitamin D, LE and LE + vitamin D treated groups), as well as a positive effect on hepatocyte ballooning. Again, in the vitamin D-enriched mushroom groups, there were significant decreases in triglyceridemia (from 103 to 75, 69 and 72 mg/dL), cholesterolaemia (from 267 to 160, 157 and 184 mg/dL), and LDL-cholesterol (from 193 mg/dL to 133, 115 and 124 mg/dL), as well as an increase in the HDL/LDL ratio and an improvement in glycaemia. Overall, these effects were correlated with a positive immunomodulation, as suggested by an increase in the CD4/CD8 lymphocyte ratio (from 1.38 in the control group to 1.69, 1.71 and 1.63 respectively in the other groups) and a switch from an increased activation of pro-inflammatory cytokines to a reduction in pro-inflammatory cytokines and a correlated increase in anti-inflammatory cytokine expression.

In the study conducted by Babu et al. (2014) [46], the effects of a 10-week supplementation of enriched mushrooms on 100 weaned Sprague–Dawley female rats (3 weeks of age, Harlan, Indianapolis, IN, USA) were studied. The rats were fed ad libitum with five different dietary patterns, all prepared according to the AIN-93G standard (Research Diets, New Brunswick, NJ, Canada); in addition, the diet of these animals was designed to provide different amounts of vitamin D3 andD2 from mushroom powder exposed to ultraviolet light or from unexposed mushroom powder. The samples yielded 15 and <0.5 mcg vitamin D2 per gram of dried mushrooms. The control and vitamin D diets contained no mushroom powder and were designed to provide intakes equal to the recommended amount of vitamin D3 or no vitamin D3 at all, whereas the other test diets were all prepared with a mixture of 5.0% unexposed mushroom powder and 2.5% or 5.0% UVB-exposed mushroom powder, respectively. From the experimenters’ estimates, it was calculated that the experimental models consumed 20 (control D3), 0 (vitD3-deficient), 2.4 (D2 from unexposed mushrooms), 300 (D2 from mushrooms exposed to 2.5% UVB light), and 600 IU of vitamin D per day (D2 from mushrooms exposed to 5% UVB light) from each of the provided diets. At the end of the experimental protocol, the animal models were individually treated with intraperitoneal saline or Escherichia coli LPS (100 mcg/kg body weight) and necropsied after 3 h, from which blood and spleen samples were obtained for subsequent analysis. From the experimental results, at 10 weeks after the start of the experiment, in none of the protocols tested with mushrooms (vitamin D-enriched and control mushrooms) was a statistically significant difference found in final body weight. Rats fed a mushroom diet exposed to 5.0% UVB had a mean blood value of 25(OH)Dtot of 155.4 ± 12.8 ng/mL. In the rats in the control group fed vitamin D3 and without mushrooms, an average 25(OH)Dtot value of 32 ± 11 ng/mL was found, which is in line with what is considered normal in humans (20–100 ng/mL), while those fed vitamin D-deficient diets had 25(OH)Dtot values of less than 10 ng/mL, values considered deficient in humans. These same rats had higher PTH levels than the groups fed a low vitamin D diet, but this did not reach statistical significance due to the wide variation in measurements within each feeding group. Furthermore, the mean PTH values of rats fed a mushroom diet not exposed to 5.0 percent vitamin D were found to be increased compared to the control group. The plasma levels of TNF-a and MIP2 were significantly lower in rats fed 2.5% and 5% UVB-exposed mushroom diets than in rats fed the control diet; however, IL-1b was significantly higher in rats fed 5% unexposed mushroom and 2.5% UVB-exposed mushrooms, suggesting that the differential increase may be related to a component of the mushroom other than vitamin D. The plasma levels of IFN-a were not altered by unexposed or exposed mushrooms. Regarding the characterization of splenocyte subpopulations, mitogenic response, and cytokine production, the percentage of CD3+, CD4+, CD8+, CD45+, and CD25+ did not differ by consumption of UVB-exposed mushrooms compared with the control diet. Functional assessment of splenic T lymphocytes with Con A and B lymphocytes with LPS showed no differences in proliferation potential between all dietary groups. In addition, dietary treatment had no effect on Con A-mediated secretion of IFN-y, TNF-y, or IL-2, suggesting that vitamin D2 from UVB-exposed mushrooms or dietary mushrooms alone is not sufficient to improve the adaptive immune response compared with the control diet. Experimental results showed an improvement in NK-cell activity in the 5% UVB-exposed mushroom group compared to the control group.

Overall, the researchers concluded from the results that an increased intake of vitamin D2 by supplementing a normal diet with vitamin Ds-enriched mushrooms may have a positive immunomodulatory effect on innate immunity. The author concluded that animals fed definitive vitamin D diets without mushrooms were characterized by a reduction in the expression of inflammatory mediators in the spleen, and this was correlated with a decrease in the availability of the precursor 25(OH)Dtot for spleen cells, without which the conversion of calcitriol (1,25-dihydroxyvitamin D), which binds to the vitamin D receptor, would decrease, inducing activation of the gene transcription of many genes related to innate immune function. Furthermore, these supplementation protocols were able to induce, in the rats studied, an increase in 25(OH)Dtot values, which in turn was associated with an improved innate immune response and a more functional immunomodulation.

In a 45-day longitudinal study (Dowley A. et al., 2021) [47] conducted in Ireland on 192 pigs (96 males and 96 females), animals fed UVB-exposed mushrooms had significantly higher plasma 25OHD levels, which was associated with enhanced innate immune responses and the induction of anti-inflammatory effects. The experimental protocol was divided into four experimental groups, all fed a standard diet but with different supplementation protocols: The first group received no supplementation at all (control group); the second group was supplemented with ZnO; the third group was treated with unenriched mushroom powder (MP, Agaricus bisporus) such that it provided an amount of β-glucans equivalent to 200 mg/kg feed; and finally, the fourth group was supplemented with a vitamin D2-enriched mushroom powder (MPD2) that provided the same amount of β-glucans as the third group but an additional vitamin D2 content of 100 µg/kg feed. The authors examined growth performance, small intestinal morphology, volatile fatty acids (VFAs), nutrient transporter gene expression, expression of genes involved in inflammation, and epithelial barrier growth performance. The MPD2-supplemented pigs had lower average daily gain (ADG) and ADFI (*p* < 0.05) compared with the ZnO and control groups. The supplemented pigs in the fourth experimental group had lower average daily gain (ADG) and ADFI (*p* < 0.05) compared with the first and second groups. Pigs in the third group (treated with mushrooms that were not enriched in vitamin D2) had lower ADG (*p* < 0,05) than the ZnO group and a lower ADFI (*p* < 0.05) than the ZnO and control groups. Again, animals in the first group had lower fecal scores (*p* < 0.05) and a lower frequency of diarrhea (*p* < 0.001) than all other groups (*p* < 0.001). Again, the experimental results showed no significant differences (*p* > 0.05) in fecal scores or diarrhea frequency between the MP, MPD2, and control groups. During fasting, the pigs in the second and fourth groups showed an increase in villus height (VH) (*p* < 0.05) compared to the control group. Pigs in the third group showed a statistically significant decrease in CD (*p* < 0.05) compared with the second group and the control group. In addition, an increase in the production of total VFA in the colon was observed in the pigs of the fourth group compared with all other groups (*p* < 0.05), and a decrease in the molar proportions of isobutyrate compared with the control group and in the molar proportions of isobutyrate, butyrate, and isovaline compared with the pigs in the second group. In pigs supplemented with MPD2, induction of the expression of SLC15A1 (*p* < 0.05) and FABP2 (*p* < 0.05) expression as well as an increased expression of the vitamin D receptor VDR (*p* < 0.05) were found in the duodenum compared with the control and ZnO groups. In the duodenum, the pigs in the fourth group also had lower expression of the chemokine CXCL8 compared with the second group and lower gene expression (*p* < 0.05) compared with the control group. In the ileum, pigs in the fourth group had higher IL10 cytokine gene expression (*p* < 0.05) compared with the control and ZnO groups. It is also interesting that in the pigs of the third group, the expression of CXCL8 (*p* < 0.05) in the ileum was lower than in the MPD2 group.

Conway E. et al. (2021) [50] conducted a study on 192 pigs for 35 days, with an experimental protocol similar to that proposed by Dowley A. et al. 2021 [41], discussed above. In terms of growth performance, pigs supplemented with MP and MPD2 had a lower ADFI at the end of the experiment (*p* < 0.05) than pigs supplemented with ZnO and the control diet, with no differences in fecal counts between the experimental groups (*p* > 0.05). Again, the MP-treated pigs had higher concentrations of total fecal VFA compared to the MPD2 and ZnO groups, and in particular, the acetate concentration was increased (*p* < 0.05) in the feces of pigs offered MP compared to those offered MPD2.

Bennett and colleagues (2013) [49] studied the effects of EM on learning and memory, possible toxicity estimated through biomarkers of liver function, brain amyloid beta (Aβ), and biomarkers of inflammation. The feed with vitamin D2-enriched mushrooms (VDM) contained 1.35 µg/kg (54 IU/kg) of vitamin D2. From the results of instrumental analyses, an increase in ergosterol and vitamin D2 of 94.5% and 100%, respectively, was found between the VDM and control feed, and a reduction in cholesterol of 5.3% in the control feed. Compared to mice in the control group, Tg mice on the VDM diet had a larger (*p* < 0.05) number of IL-10-positive neurons in the cortex and a significantly (*p* < 0.01) larger area of IL-10-positive neurons. On the other hand, no significant differences were found between genotype and feeding type in the immunolocalization levels of IL-1β in the cortex or hippocampus of brain sections; but a significant effect of feeding on the total area of neurons in the cortex was detected. The absolute sensitivity of IL-1β staining and the size of the total area of neurons were lower than those of IL-10. In conclusion, wild-type and AD transgenic mice fed with vitamin D2-enriched mushrooms (VDM) showed a general improvement in memory and learning abilities, as well as a decrease in amyloid plaque load and glial fibrillary acidic protein and an increase in brain IL-10 expression, compared to the control groups.

Malik et al. (2022) [48] conducted a 4-week longitudinal study in India on 36 albino Wistar rats to investigate the effects of the treatment on blood biochemical parameters and bone homeostasis, as well as on the expression levels of bound vitamin D (CYP2R1, CYP27B1 and VDR) in the liver and kidneys of the rats. Experimental protocols with shiitake mushrooms, buttons, and oysters were designed to provide approximately 30 IU/day of vitamin D. Analysis of the experimental results did not reveal a statistically significant effect on body weight in the different experimental groups; but at the end of the study, there were positive and significant (*p* < 0.05) changes in 25(OH)Dtot concentrations compared to the baseline for the GP-2 treatment groups (11.68 ± 1.92 to 46.00) ± 7, 61 ng/mL), GP-3 (from 16.92 ± 0.48 to 49.96 ± 5.42 ng/mL), GP-4 (from 15.32 ± 1.28 to 43.62 ± 5.83 ng/mL), GP-5 (from 12.48 ± 2.12 to 55.14 ± 6.60 ng/mL), and GP-6 (from 14.22 ± 2.49 to 66.14 ± 6.32 ng/mL). Again, the experimental results revealed a significant (*p* < 0.05) increase in calcemia and phosphatemia in the treated groups as well as a general decrease in alkaline phosphatase and parathyroid hormone levels. Again, on microscopic examination, mild osteoporotic changes were evident, with large, separated trabeculae and reduced cortical thickness in the GP-1 group, in contrast to the mushroom-treated GP-2, GP-3, and GP-4 groups and in the GP-5 and GP-6 groups supplemented with non-fungal vitamin D. The results showed a significant increase in raft cleavage and a significant decrease in bone area in the selected region of interest compared to the control group (*p* < 0.05). In the mushroom group, an upregulation of CYP2R1 (Ct = 27.56), as well as of the VDR, was found to be 2.47-fold in the vitamin D-enriched mushroom group; a value not different from that found in the vitamin D3-treated group (2.5-fold). Again, experimental results showed a downregulation of hepatic CYP27B1. In the kidney, VDR mRNA expression was found to be increased in both the mushroom- and vitamin D2/D3-supplemented groups, although this increase was greater in the vitamin D2 and D3-supplemented groups than in the mushroom-treated group (~50-fold and ~24-fold, respectively). Again, the researchers reported a ~1.8-, 6.8-, and 1.5-fold increase in renal CYP27B1 expression in the shiitake, champignon mushroom, and oyster mushroom-treated groups (GP-2, GP-3 and GP-4), respectively, compared to the control group (GP-1). CYP27B1 expression levels were approximately similar in the vitamin D2-fed group (GP-5), and lower in the vitamin D3-fed group (GP). In conclusion, the authors of the study suggest that vitamin D2 obtained from mushrooms irradiated with UVB light is bioavailable and functional in maintaining bone health and mineralization in a mouse model without obvious adverse effects. Furthermore, this supplementation protocol exerts induction effects on the expression of VDR, which is found to be highly expressed in both liver and kidney, while CYP2R1 is more highly expressed in the liver and CYP27B1 is prevalent in the kidney.

## 4. Discussion

### 4.1. Classical and Nonclassical Function of Vitamin D in Human Studies

In regard to human samples, results of the studies investigated concerned mainly the nonclassical functions, while the classical effects concerned only EM effects on serum vitamin D levels (serum 25(OH)D2, serum 25(OH)D23 and 25(OH)Dtot).

The nonclassical functions regarded different outcomes ranging from muscle damage, DOMS, and inflammation pertaining to cognitive and immune system functions. Generally, studies in humans have not reported significant effects on 25(OH)Dtot values, which is probably why no positive effects have been found on most of the outcomes related to the nonclassical vitamin D functions, such as neuromuscular pain, cognitive function, and mood.

Concerning the effects of metabolic syndrome markers, two studies were conducted on two very different samples, duration, protocols, and posology (vitamin D dosage). Indeed, results of the study including 36 overweight (BMI > 25 kg/m^2^), pre-diabetic and vitamin D-deficient adults (25(OH)Dtot ≤ 20 ng/mL) yielded no positive and significant changes with any treatment over time [39]. Conversely, the protocol led on older healthy participants (45–65 years old) highlighted a significant decrease in plasminogen activator inhibitor-1 (PAI-1), a risk marker for coronary artery disease. The authors pointed out that such results have been observed in a limited healthy population and for a limited period of time, so this should be confirmed by future research. Moreover, they also underlined that it has to be clearly investigated if the above mentioned change is actually attributable to modifications in 25(OH)D2 levels or to other bioactives present/intrinsic in the mushrooms [42].

### 4.2. Classical and Nonclassical Function of Vitamin D in Animal Studies

In regard to animal samples, the results of the studies investigated concerned mainly the classical functions, while the nonclassical effects concerned the immune system, microbiology, volatile fatty acids, coefficient of apparent total tract digestibility (CATTD), gastrointestinal morphology, and nutrient transporter genes.

The nonclassical functions of the immune system were analyzed in three studies in animal models [46,47,49]. Overall, the studies suggest that the supplementation protocols may exert positive effects on innate immunity through enhanced NK-cell activity, a positive modulation in an anti-inflammatory direction, of certain chemokines and cytokines such as IL10, CXCL8 and IL6, in the duodenum and ileum in response to stimulation with LPS [46,47]. Interestingly, the same signaling was elevated in the brains of active samples involved in the study by Bennett and colleagues (2013) [49].

The results of Conway and colleagues (2021) [50] and Dowley et al. (2021) [47] showed positive effects of the treatments on the amount of volatile fatty acids (VFA) produced by the gut flora, an improvement in VH (villus height) in the duodenum and jejunum in the animal models tested and an increase in the expression of SLC15A1 (solute transporter family) and FABP2 (fatty acid binding protein) in the duodenum compared to the control group.

The classical effects in animal models showed important and overall positive results. In the case of animal models, in all studies analyzed, regardless of treatment, animal sample or protocol, 25(OH)D levels were significantly increased compared to controls, as well as a significant reduction in PTH levels [44,45,48] in the studies in which it was analyzed. Notably, in a study conducted by Calvo et al. (2012) [44], this result was also evident in rats that consumed unexposed mushrooms compared to the control group, suggesting that the mushrooms themselves may contain factors that facilitate calcium absorption and thus reduce PTH secretion. With regard to calcium levels, only three studies showed results in this direction [45,48,51], all pointing out that Ca++ levels increased after treatment. Still, the study Lee et al. (2009) [51], reported increased mRNA levels of active calcium transport genes (duodenal CABP9K, TRPV6, and renal CABP9K, TRPV5,6). Again, Malik et al. (2022) [48] found increased expression of VDR in the liver and kidney after treatment, while Dowley et al. (2021) [47] showed increased expression in the duodenum.

With regard to bone density and morphology, all animal studies included in this review reported positive results. In particular, Calvo et al. (2012) [44] showed an improvement in bone growth (particularly of the femur), Jasinghe et al. (2006) [45] an increase in BMD and bone length of the femur, Malik et al. (2022) [48] a significant reduction in osteoid area, and Lee et al. (2009) [51] showed beneficial effects on bone density and bone length morphology.

Specifically, two studies focused on the effects of EM on female mice undergoing sham operation/oophorectomy; and in both studies, the treatment showed significant beneficial effects in the absence of estrogen [52,53], probably due to both its polyphenol and fiber content and its high vitamin D2 content. Finally, Drori and colleagues (2016) [54] showed positive effects of EM in insulin damage and resistance (NAFLD). Furthermore, a synergistic effect on body fat accumulation was observed, so overall, the results seem to support the potential use of EM in patients with early-stage NASH.

Finally, it is interesting to note that in the study by Duffy and colleagues (2018) [55], it was reported that treatment with EM mushrooms and vitamin D₃ delayed lipid peroxidation of meat samples compared to vitamin D3 and vitamin D2 treatments.

### 4.3. Comparison of Animal and Human Studies

Overall, studies conducted on humans and animal models are heterogeneous, both with regard to dosage, treatment times, and samples analyzed. Some human studies were conducted on particular samples, such as professional athletes [41,43] or elderly subjects [38,42]. Similarly, treatment times and treatment modalities were different in studies conducted in animal models, even when grouping studies on the same species. Furthermore, in animal models, some studies included the combined treatment of vitamin D-enriched mushrooms and other substances. Despite the heterogeneity of the studies reported, the evidence gathered in this review suggests overall that the intake of vitamin D2-enriched mushrooms has substantially different effects in animal models and in human studies.

In contrast to the results in humans, studies in animal models report that treatment with these fortified foods generally induces a positive effect on clinical outcomes related to the classical and nonclassical vitamin D functions, but it is noteworthy that studies in animal models have longer treatment times relative to the average lifespan and significantly higher dosages of vitamin D intake per kilogram of body weight than studies in humans. Furthermore, intervention studies in animals generally induce greater increases in circulating levels of 25(OH)Dtot than studies in humans. Although it is not technically correct to directly compare the results obtained in animal models with those obtained in human subjects, the presence of overall positive results in all the different species analyzed (e.g., rat vs. pig) and the higher levels of 25OHDtot achieved in animal model studies suggest that the longer time and higher qualitative levels of dietary supplemented D2 could at least partially justify the difference in clinical results between animal model and human studies. According to some work [9,30], vitamin D3 probably induces a more rapid and sustained increase in 25(OH)Dtot concentration than vitamin D2, and this phenomenon could be attributed to the structural differences of the two isoforms described above. In particular, the additional double bond and methyl group present in vitamin D2 would influence the rate of vitamin D hydroxylation. Indeed, vitamin D3 is considered a preferential substrate for hepatic 25-hydroxylase and has a higher binding affinity to vitamin D-binding proteins, which are responsible for vitamin D transport through the circulatory system [23]. Thus, pre-vitamin D3 (25(OH)D3) is a more tightly bound substrate to the 25-hydroxylase enzyme than 25(OH)D2, and this difference induces significant cellular effects. Consequently, higher dosages may be required to induce significant effects on final clinical outcomes related to vitamin D activity. Furthermore, it cannot be excluded that longer treatment times have the potential to induce increased vitamin D2 activity. Indeed, cytochrome CYP3A1, which is characterized by vitamin D degradation activity, has a more rapid activity on 1a25OH2D2 than on 1a25OH2D3, suggesting that this enzyme might limit the action of vitamin D2 at the cellular level, and this might partly justify the evidence that the D2 isoform generally induces fewer clinical effects than the D3 isoform [23,43].

The dosages of enriched mushrooms to be used in studies should be such that they induce increases in 25(OH)Dtot concentrations that are certainly greater and closer to those obtained in animal models. However, it should be emphasized that values above 100 ng/mL could be dangerous in humans. Consequently, dosages should be such as to induce increases in total vitamin D concentrations that pose a health risk to test subjects.

Furthermore, considering that prolonged exposure times could also play a key role, it might also be useful to assess the effect of the time variable on the final clinical results in intervention studies; this effect could be evaluated by comparing groups treated for different times but with comparable 25(OH)Dtot values. Furthermore, this hypothesis could be further strengthened by assessing the presence of possible epidemiological correlations between clinical outcomes related to the classical and nonclassical vitamin D functions and mushroom intake, as well as by intervention studies on humans with longer exposure times than those studied so far.

Finally, it is necessary to investigate the effects of other variables at play, such as the time of exposure to ultraviolet rays, the ideal conditions of vitamin D2 conversion in the food matrix for the 10 commercial mushroom species analyzed, and possible differences between whole, freeze-dried, and/or powdered foods with the same VitD values per kg of food consumed. This knowledge could be very important for the effective implementation of biofortification policies for these foods, which could be very useful in some particular cases, such as in vegans who are exposed to little sunlight, as well as in the use of vitamin D-enriched mushrooms in the dietary-therapeutic treatment of conditions such as osteopenia or osteoporosis.

At present, therefore, the implementation of enriched mushrooms in the development of omnivorous diets or food pyramids remains controversial, as there is a risk of a potential paradox effect that could negatively influence vitamin D-related clinical outcomes. However, data obtained in the future might justify the long-term use of these foods in particular individuals, such as vegans or those allergic to fish protein.

## 5. Conclusions

The aim of this systematic review was to summarize what has been reported in the literature on the treatment of animal and human models with irradiated commercial mushrooms, with particular regard to the effects of treatment on clinical outcomes associated with the classical and nonclassical vitamin D functions.

The results of this systematic review indicate that the intake of vitamin D from irradiated mushrooms could be a viable alternative for meeting vitamin D requirements, since in humans they induce an increase in blood values of vitamin D2 anyway, and in the animal model they are also effective in increasing blood values of 25(OH)Dtot, which is likely to result in clinical efficacy on clinical outcomes related to vitamin D function. However, the dosage and probably the treatment time tested in humans should be reviewed. Hence, clinical studies conducted over longer periods and with generally higher and more defined dosages are required before EM consumption can actually be considered as a real source of vitamin D, in order to definitively assess whether the clinical efficacy shown so far only in the animal model for both the classical (bone metabolism) and nonclassical (muscle function, cognitive performance, anti-inflammatory and antioxidant activities) effects of vitamin D can also be verified in humans.

## Figures and Tables

**Figure 1 antioxidants-12-00736-f001:**
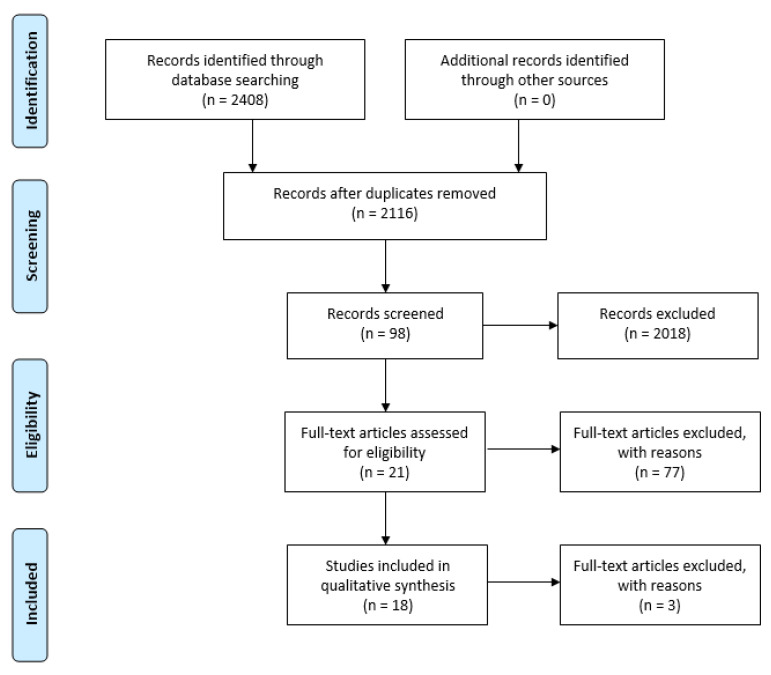
PRISMA Flowchart of study selection process [37].

**Table 1 antioxidants-12-00736-t001:** Technological and environmental factors related to the vitamin D2 improvement in edible mushrooms.

Technological and Environmental Factors	Comment	Reference
Type of effective irradiation	Increased effectiveness using UVB irradiation and sun exposure, rather than UVA and UVC irradiation.	[1,23,24,25,26,27,28,29,36]
Irradiation surface	The smaller the sample, the larger the exposure area and thus the greater the conversion of vitamin D2. Again, irradiating all sides of the cut allows a greater synthesis of vitamin D2 than simply exposing only one side.	[1,23,24,25]
Ideal reaction temperature	Between 25–29 °C. Data still insufficient and preliminary.	[1,23,24,25,36]
Exposure intensity	Higher efficacy for values between 1.14 and 1.36 W/m^2^ (values referring exclusively to irradiation with pulsed rays). There are no data correlating the conversion of ergosterols to ergocalciferol with the environmental UV index and/or factors such as latitude and time of day. Data still insufficient and preliminary.	[1,23,24,25]
Product type	Higher conversion efficiency in freeze-dried and heat-dried mushrooms than in fresh mushrooms. Data still insufficient and preliminary.	[1,23,25,36]
Sample storage	In fresh mushrooms, loss of about 23% vitamin D2 in mushrooms stored at 3–2 °C due to sample degradation, but no significant loss for storage at 4 °C. At room temperature, dried samples appear to have losses of about 50% after 18 months. Still insufficient and preliminary data.	[1,23,24,25]
Culinary transformations	Cooking induces a loss of vitamin D2, which differs depending on the type of cooking (greater in baking than in pan-frying). Data still insufficient and preliminary.	[1,23]

**Table 2 antioxidants-12-00736-t002:** Studies on human samples included in the systematic review.

Subjects	Reaction Conditions and Sample Type	Vitamin D2 EM Supplementation and Species	Vitamin D Nonclassic Effects	Vitamin D Classic Effects	Reference
N = 436 (healthy, ≥60 years old, M-F)	6 months freeze-dried, powder and capsuled, UVB Lamp	600 IU/die, Agaricus bisporus	No benefit on cognitive functions and mood	Increase in 25(OH)D2. Decline in total 25(OHD). Such decline was observed to be less negative and slower in the D2 arm (EM) than controls.	[38]
N = 36 (pre-diabetic, BMI < 25, vitamin D-deficient adults, 49 ± 12 years old, M-F)	4 months fresh sliced cooked mushrooms, UVB Lamp	arm 1: 600 IU/die arm 2: 4000 IU/die Agaricus bisporus	No positive or significant results on metabolic syndrome markers	Modest, but significant increase in serum 25(OH)D2 over time. The amount of 25(OH)D3 and total 25(OH)D remained unchanged.	[39]
N = 40 (healthy adults, 20–50 years old, M-F)	6 weeks fresh sliced cooked mushrooms cooked mushrooms, UVB Lamp	25 µg ergocalciferol/die arm 1: 8.8 µg/die arm 2: 17.1 µg/die Agaricus bisporus	Not studied by the authors	Increase in serum 25(OH)D2 in both arms, which was correlated to the treatment. No influence of any arm treatment on total 25(OH)D while 25(OH)D3 levels declined.	[40]
N = 33 (athletes students with serum 25(OH)D less than 30 ng/mL, 16.2 ± 0.19 years old)	6 weeks powder and capsuled, UVB Lamp	600 IU/die Agaricus bisporus	No positive results muscle system/function, exercise-induced muscle damage or DOMS	Increase in either 25(OH)D2 or total 25(OH)D. Levels of 25(OH)D3 decreased.	[41]
N = 90 (healthy adults 40–65 years old, M-F)	4 weeks Freeze-dried, powder, UVB Lamp	600 IU/die Agaricus bisporus	Significant decrease in Plasminogen activator inhibitor-1 (PAI-1)	No significant changes in total 25(OH)D and 25(OH)D3 values. Increase in 25(OH)D2 levels.	[42]
N = 28 (healthy athlete professional pilots, age not declared, M)	6 weeks powdered, hot dried, UVB Lamp	3800 IU/die Agaricus bisporus	No influence on muscle system/function, exercise-induced muscle damage or DOMS and negative effects on their markers such as CK, LDH, serum myoglobin and DOMS as well as amplified exercise-induced muscle damage (EIMD)	No significant change in total 25(OH)D values, but significant increase in serum 25(OH)D2 levels and a significant decrease in serum 25(OH)D3 levels.	[43]

**Table 3 antioxidants-12-00736-t003:** Studies on animal samples included in the systematic review.

Subjects	Reaction Conditions and Sample Type	Vitamin D2 EM Supplementation and Species	Vitamin D Nonclassic Effects	Vitamin D Classic Effects	Reference
300 rats 3 weeks ol—F	10 weeks powdered, freeze-dried, UVB Lamp	arm 1: 300 IU/day arm 2: 600 IU/day Agaricus bisporus		Vitamin D2 from mushrooms was bioavailable and effective in suppressing PTH levels. The mean PTH level in the group of rats fed with the 5.0% of unexposed mushroom diet was also significantly lower (*p* < 0.002) than in the vitamin D-deficient fed control group. Rats fed the 5.0% UVB mushroom diet had a mean 25(OH)D plasma level of 159 ± 29 ng/mL, while the control group had a mean 25(OH)Dtot final level of 32 ± 11 ng/mL. The tested treatment, appeared to stimulate bone formation in all of the experimental group, inducing significantly longer femurs, *p* < 0.001), and a positive effect on cortical bone (The treatment also had a positive effect on cortical bone (higher mean midshaft cortical thickness and pMOI, *p* < 0.01).	[44]
30 rats (age not declared—M)	4 weeks powdered, hot dried, UVB Lamp	1 mg/die Lentinula edodes		The serum 25OHD concentration of the active group was 129 ± 42 (SD 22 ± 00) nmol/L in contrast to the control group (6 ± 06) (SD 1 ± 09) nmol/L. The serum calcium level of the active group was significantly lower compared to the controls. Significantly higher BMD and femur lengths in the active group. Decreased serum PTH levels, increased serum ionized Ca levels and an age-related decrease in duodenal Ca absorption have been previously reported.	[45]
100 rats (3 wk old)	10 weeks Powdered, freeze-dried, UVB Lamp	arm 1: 300 IU/die arm 2: 600 IU/die Agaricus bisporus	Plasma TNF-a and MIP2 were significantly lower in rats fed 2.5 and 5% UVB-irradiated mushrooms compared with controls; IL-1b was significantly higher in rats fed 5% non-irradiated and 2.5% UVB-irradiated mushrooms. NK-cell activity was improved in the 5% UVB-exposed mushroom group compared to controls. UVB-exposed mushrooms in the diet caused a significant reduction in chemokines and cytokines in response to LPS stimulation.	Rats fed 5.0% UVB-exposed mushroom diet (600 IU) had a mean plasma 25(OH)Dtot (155.4 ± 12.8 ng/mL) in respect of the control.	[46]
192 pigs (28 days old, 96 M and 96 F)	45 days dried powder, UVB lamp	100 ug/kg/feed Agaricus bisporus	Concerning gastrointestinal morphology, treatment led to a significant improvement of VH (villus height). Pigs supplemented with MPD2 had increased total colonic VFA’s compared with all other groups (*p* < 0.05). Treatment also increased expression of SLC15A1, (*p* < 0.05) and FABP2, (*p* < 0.05) in the duodenum compared with the control group. In the duodenum, treatment caused a reduction in the expression of chemokine CXCL8, (*p* < 0.05) compared with the ZnO group and a reduction in the expression of cytokine gene IL6, (*p* < 0.05) compared to the control group. In the ileum, treatment led to an increase in the expression of cytokine gene IL10, (*p* < 0.05) compared with the control and ZnO group.	Increased expression of vitamin D receptor VDR, (*p* < 0.05) in the duodenum compared to the control and ZnO group.	[47]
36 rats (4–6 weeks old, M-F)	4 weeks powdered, hot dried, UVB Lamp	30 IU/die Lentinula edodes Pleurotus ostreatus Agaricus bisporus		Treatment caused a significant increase in 25(OH)D. A significant increase (*p* < 0.05) was seen in calcium and phosphorus levels, while a significant decrease in alkaline phosphatase and PTH levels was seen in all treated groups. The results showed a significant increase in trabecular separation and a significant decrease in osteoid area in the selected region of interest in relation to the control group (*p* < 0.05).	[48]
21 wild-type (B6C3) and 25 transgenic (APPSwe/ PS1dE9) mice (2 months old, gender not declared)	7 months powdered, freeze-dried, UV-C Lamp	54 IU/Kg/die Agaricus bisporus	Treatment resulted in significantly (*p* < 0.05) higher number of IL-10-positive neurons in the cortex and a significantly (*p* < 0.01) larger area of neurons was IL-10 positive. Immunolocalization of IL-1β in the irradiation cortex or hippocampus of 9-month-old mice showed no difference between chow type or genotype, but a significant main effect of chow on the total area of neurons in the cortex. The absolute sensitivity of IL-1β staining and the size of the total neuronal area appeared to be lower than that of IL-10, based on the comparison of the stained areas. In conclusion, VDM-fed wild-type and AD transgenic mice showed improved learning and memory performance, significantly reduced amyloid plaque load and glial fibrillary acidic protein, and increased brain interleukin-10 concentration. The results suggest that VDM may be a dietary source of vitamin D2 and other bioactives to prevent memory impairment in dementia.	Treatment led to an increase in vitamin D2.	[49]
192 pigs (age not declared, M-F)	35 days dried powder, UVB Lamp	100 ug/kg/feed Agaricus bisporus	No effect of treatment on the number of total bacteria. Pigs in the control group had higher concentrations of total fecal VFA than the treated group. With regard to the coefficient of apparent total tract digestibility (CATTD), the authors found no differences between the groups.		[50]
55 mice (3 weeks old, M)	4 weeks Freeze-dried, powder, UVB Lamp	1 ug/die Active treatment: Ca^++^ and Vitamin D2 EM Lentinula edodes		Beneficial effects of Ca^++^-EM treatment on serum calcium levels, mRNA levels of active calcium transport genes (duodenal CABP9K, TRPV6 and renal CABP9K, TRPV5,6) and femur density and length bone histology.	[51]
48 rats (subjected to sham operation or bilateral ovariectomy, when they were 5 weeks old, F)	6 weeks Powder, Hot dried, UVB lamp	750 ug/KgDiet 30 g powder/die Lentinula edodes		Comparison of 25(OH)D2 levels between the sham UV(O) and OVX-UV (O) groups showed that its level decreased by 70% in the absence of estrogen, suggesting that the presence of ovaries is highly associated with vitamin D2 bioavailability. The absence of estrogen had negative effects on trabecular bone structure and the bioavailability of vitamin D and calcium. Although there was a decrease in calcium levels, serum 25(OH)Dtot and other parameters linked to trabecular bone structures including BMD, vitamin D2- EM. These results may help delay bone loss which can be accelerated by the absence of estrogens after menopause.	[52]
32 mice (some of them subjected to sham operation or bilateral ovariectomy,7 weeks old, F)	23 weeks Powdered, Freeze-dried, UVC Lamp	5 ug/die Pleurotus eryngii		Pulsed enhanced vitamin D2 EM can maintain bone health, decrease the activity of bone resorption markers (osteocalcin, PYD, NTX1) and increase bone health related metabolites (osteocalcin, taurine, cretainin, emaic lactate, arginine) in OVX mice through its action as a mushroom (polyphenols and fiber) and its vitamin D2 content.	[53]
Four groups of mice (*n* = 5/6) mice per group. Concanavalin A induced immune liver damage)	25 weeks	1 IU of vitamin D twice daily for 3 days Lentinula edodes	Beneficial effects of EM on liver damage and insulin resistance in a mouse model of NAFLD. A synergistic effect on body fat accumulation was observed. Given the high safety profile of these extracts, the data support their potential use in early-stage patients NASH.		[54]
120 pigs (60 M ad 60 F, age not declared)	DC	55 days Freeze-dried, powder, UVB Lamp	50 μg of vitamin D₂/kg/feed Study focused on the organoleptic quality of the meat (not reported but one from Monaghan Mushrooms that sell only commercial mushrooms)		[55]

## Data Availability

Not applicable.

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
