# Peer review of "Vitamin D from UV-Irradiated Mushrooms as a Way for Vitamin D Supplementation: A Systematic Review on Classic and Nonclassic Effects in Human and Animal Models"

_antioxidants, 2023, doi:10.3390/antiox12030736_

Round 1

Reviewer 1 Report

The article deals with an interesting topic, because plant-based diets, in which there are almost no sources of vitamin D, are becoming more and more popular. Taking up such an issue is very right.

Comments:

 First of all, at the beginning, I must point out that the page numbering in the article is doubled, which causes a huge problem when reading the article after printing it in paper version. It takes a lot of work to put the pages in order.

- in Abstract it is stated that there were 13 studies on animals, while in the whole article and in table 3 there are 12 articles

In the Introduction, the sentence "Ergocalciferol (vitamin D2) is the most widely used vitamer in fortified foods and dietary supplements" is questionable. In my opinion, it is quite the opposite, vitamin D3 is most often used. This is the case, for example, in Poland, but also in Italy [Moretti M. et al.: Italian Public Health Expenditure for Vitamin D is still high: New Outlook to Saving From a Consumption Analysis in the Liguria Region. Clin Ther. 2021, 43(11): 1969-1982].

Bovine liver is not a good source of vitamin D, eggs are much better.

Since the authors write that mushrooms are naturally rich in vitamin D2, it is necessary to provide literature data on how much of this vitamin is in mushrooms.  

In Results In the description of the study, ZajÄ…c et al. groups of test persons are confused. The second group was a group of people who were given vitamin D3, while in the text of the article the authors write twice that it was the third group?

In addition, once the authors write "slight increase in 25(OH)D3 levels in the third experimental group, i.e. the vitamin D3-treated group (p<0.001)” and once “Finally, 25(OH)D3 levels were unchanged in the vitamin D3-treated subjects? This is incomprehensible

Discussing the study by Mehrotra A et al. the authors write that it was carried out on a group of 43 prediabetic adults with BMI >25, while in table 2 there is "N=36 pre-diabetic BMI <25" ??

Table 2 in the study by Stephensen et al. 25 mg of ergocalciferol is given. However, in the original publication of this study, it is 25 µg (the sum of 0.85 µg and 17.1 µg). There are errors in vitamin D units throughout this paragraph !!

Table 3 should be titled "Studies on animal samples" and not "Studies on human samples"

In the description of animal studies, the study of Lee et al. is missing from the text of the article.

 In the description of the study, Nieman et al. a group of 30 US athletes is given, and table 3 shows N=28

Instead of "significant increase in serum 25(OH)D2 levels to 9.3 ng/mL" in the description of the Shanely study, shouldn't there be "significant increase in serum 25(OH)D2 levels of 9.3 ng/mL" ?

I would suggest shortening the description of some studies, e.g. Babu et al.

The abbreviations BMD, MB, AST, CPK are not explained. The abbreviation DC is not explained in table 2.

 25OHD, 25(OH)D, 25(OH)D2, 25(OH)D3, 25-OH-D2, 25(OH)Dtot, 25OHD3, 25OHD2 - spelling should be corrected

Reviewer 2 Report

The authors have conducted a systematic review on the classical and non-classical functions of  vitamin D associated with clinical effects in the treatments of animal and human models with irradiated commercial mushrooms.  In a total of 19 studies, the authors reviewed that the intake of vitamin D from irradiated mushrooms could possibly help meet vitamin D needs, but the dosage and the time of treatment need further evaluation, since current standards are certainly not effective in achieving a positive effect.

I have no technical concerns but I strongly believe that the overall presentation of this manuscript needs to be improved, as I have detailed below. I believe that these revisions would help improve the manuscript if a revision is requested by the editor.

Title:

remove or replace “interesting” from the title, it is hard to know what this “interesting” means.

Abstract:

The sentence “”since current standards are certainly not effective in achieving a positive effect” is confusing, how could standards be effective or not effective...?

I see some numbers are inconsistent…A total of 19 articles, but in the first sentence of Section 3. Results, 18 articles; again 19 articles, 6 on human and 13 on animal models, but then, 6-7 lines below, 9 out 12 studies on animals, …

Introduction:

Table 1, I would suggest that the authors list the taxa of the mushrooms in the Comments column so that the reader could know these magic mushrooms immediately without check the referecnes. Replace “Paper” with “Reference”

Table 1: in the caption, “enrichment”

Materials and Methods:

Instead of listed 1 and 2, the information to exclude the studies should be described in the text..

Figure 1: do you think the arrowed line leading to the box with “n = 18” is kind of too long in comparison with other arrowed lines? Also, there is an extra mark on the box with “n = 0” and move the top portion of “PRISMA 2009 Flow Diagram” or its information to the caption of this figure.

Results:

Table 2, the caption is oversimplified, should provide more details of this table; replace “Paper” with “Reference” and move the column to the end of this table, simplify this column with only number of references but not the authors and years (this is also the problem in many places in the text, repetitive citations…). Again, make sure the names of the mushrooms are provided as well.

Table 3: many of the same issues for Table 2 are also here…please revise.

Discussion:

The first two short paragraphs are repetitive, not necessary.

I highly recommend that the authors establish a few main subsections in Discussion so that the authors could re-organize numerous short paragraphs to a few concisely discussed sections.

Conclusion:

Most of the information is repeated from Results, Conclusion could be shortened by half.

Round 2

Reviewer 2 Report

I appreciate very much the authors' efforts devoted to improving their manuscript. I have no more questions.